# Initial Use Experience of Durvalumab Plus Gemcitabine and Cisplatin for Advanced Biliary Tract Cancer in a Japanese Territory Center

**DOI:** 10.3390/cancers17020314

**Published:** 2025-01-19

**Authors:** Kento Shionoya, Atsushi Sofuni, Shuntaro Mukai, Yoshiya Yamauchi, Takayoshi Tsuchiya, Reina Tanaka, Ryosuke Tonozuka, Kenjiro Yamamoto, Kazumasa Nagai, Yukitoshi Matsunami, Hiroyuki Kojima, Hirohito Minami, Noriyuki Hirakawa, Qiang Zhan, Takao Itoi

**Affiliations:** 1Department of Gastroenterology and Hepatology, Tokyo Medical University, Tokyo 160-0023, Japan; kjscf034@tokyo-med.ac.jp (K.S.); a-sofuni@tokyo-med.ac.jp (A.S.); s-mukai@tokyo-med.ac.jp (S.M.); yoyama@tokyo-med.ac.jp (Y.Y.); tsuchiya@tokyo-med.ac.jp (T.T.); r-tanaka@tokyo-med.ac.jp (R.T.); tonozuka@tokyo-med.ac.jp (R.T.); kenjiro@tokyo-med.ac.jp (K.Y.); kazu4439@tokyo-med.ac.jp (K.N.); ym1228@tokyo-med.ac.jp (Y.M.); kojima-h@tokyo-med.ac.jp (H.K.); minami@tokyo-med.ac.jp (H.M.); d121037@tokyo-med.ac.jp (N.H.); zhanq33@163.com (Q.Z.); 2Department of Clinical Oncology, Tokyo Medical University, Tokyo 160-0023, Japan; 3Departments of Gastroenterology, Wuxi People’s Hospital of Nanjing Medical University, Wuxi 214023, China

**Keywords:** bile duct cancer, chemotherapy, durvalumab, endoscopic drainage

## Abstract

Biliary tract cancers (BTCs) present a challenging prognosis. However, advancements in chemotherapy, particularly the use of targeted therapies for specific gene mutations, have led to improved patient outcomes. This retrospective single-center study evaluated 44 patients who received gemcitabine and cisplatin, combined with durvalumab (GCD) chemotherapy, from January 2023 to March 2024, with a median follow-up of 10 months. This study assessed overall survival (OS), progression-free survival (PFS), response rates, and adverse events (AEs). Results indicate an overall response rate of 23% and a disease control rate of 82%. Median OS and PFS were 15.3 and 8.0 months, respectively. Patients receiving primary chemotherapy exhibited prolonged survival, while bile duct drainage was associated with improved outcomes. Notably, Grade 3 or higher AEs occurred in 54.5% of patients. Overall, GCD chemotherapy demonstrates significant potential for BTC treatment, underscoring the importance of bile duct management in enhancing patient outcomes.

## 1. Introduction

Biliary tract cancers (BTCs), such as gallbladder cancer, intrahepatic bile duct cancer, and cholangiocarcinoma, are high-grade cancers with poor prognosis, along with pancreatic cancer [1]. However, recent advances in chemotherapy and genomics have improved the prognosis of these cancers. Particularly, compared to other gastrointestinal cancers, the percentage of mutations that can be treated is high. Although molecular-targeted drugs have been developed for gene abnormalities that are represented by isocitrate dehydrogenase (IDH)- 1/2 gene abnormalities and fibroblast growth factor receptor (FGFR)- 2 fusion genes, the number of drugs that can be used clinically is limited [2]. Gemcitabine plus cisplatin (GC) chemotherapy has been the standard of care for the primary treatment of unresectable or recurrent BTCs, but secondary treatment had not been established until recently. In Japan, tegafur/gimeracil/oteracil (S-1) has been used as a secondary treatment because it is approved by insurance. Therefore, gemcitabine plus S-1 (GS) chemotherapy and gemcitabine plus cisplatin plus S-1 (GCS) chemotherapy are also used in Japan [3,4]. In 2022, a Phase 3 study named the Topaz-1 trial showed that durvalumab had an add-on effect on GC chemotherapy. GC plus durvalumab (GCD) chemotherapy, which has been used as the conventional standard of care, and immune checkpoint inhibitors began to be used as the primary chemotherapy for BTCs [5]. However, it is a novel regimen with little experience of use and few reports; therefore, it is important to accumulate evidence. This study retrospectively examined the initial use of GCD chemotherapy for BTCs.

## 2. Materials and Methods

### 2.1. Patients

This retrospective cohort study analyzed the data of 44 consecutive patients who received Topaz-1 chemotherapy for BTCs between January 2023 and March 2024 at a single territory care center. At data cutoff (30 September 2024), the median duration of follow-up was 10 months (range 0.7–22.6). The inclusion criteria were as follows: (1) age of 20 years and older, (2) receiving GCD chemotherapy for BTCs, and (3) Eastern Cooperative Oncology Group performance status (ECOG PS) 0 to 2. The exclusion criteria were as follows: (1) age of <20 years, (2) ECOG PS 3 or more, (3) patients with significant renal or hepatic dysfunction, and (4) patients expressed a desire not to participate in this study.

### 2.2. Intervention

GCD chemotherapy was administered along with the Topaz-1 regimen. Intravenous administration was performed in a 21-day cycle for up to 8 cycles. Durvalumab (1500 mg) was administered on Day 1 of each cycle, in combination with gemcitabine (1000 mg/m^2^) and cisplatin (25 mg/m^2^), which were administered on Days 1 and 8 of each cycle. After completion of GC chemotherapy, 1500 mg of durvalumab monotherapy was administered once every 4 weeks until clinical or imaging (per RECIST v1.1) disease progression or unacceptable adverse events (AEs) took place (Figure 1) [5]. For patients with renal dysfunction or who had received cisplatin prior to GCD chemotherapy, the dose of cisplatin was reduced from the initial dose. Thereafter, the doses of gemcitabine and cisplatin were reduced according to the progress of the patients. During the period of GCD administration, blood tests for endocrinology, such as thyroid and adrenal functions, and blood tests for fibrosis markers, such as krebs von den lungen-6, were performed once a month for monitoring immune-mediated (im)-AEs. Patients with bile duct stenting for obstructive jaundice underwent endoscopic treatment and bile duct stent replacement at around three months.

GCD chemotherapy was administered along with the Topaz-1 regimen [5]. Intravenous administration was performed in a 21-day cycle for up to 8 cycles. Durvalumab (1500 mg/time) was administered on Day 1 of each cycle in combination with gemcitabine (1000 mg/m^2^) and cisplatin (25 mg/m^2^), which were administered on Days 1 and 8, respectively, of each cycle. After completion of GC chemotherapy, 1500 mg of durvalumab monotherapy was administered once every 4 weeks until clinical or imaging (per RECIST v1.1) disease progression or unacceptable adverse events.

### 2.3. Outcomes and Definitions

The study outcomes included the following: (1) overall survival (OS), (2) progression-free survival (PFS), (3) response rate, and (4) AEs and im-AEs.

OS was defined as the time from the date of induction of GCD chemotherapy to death due to any cause, while PFS was defined as the time from the date of induction of GCD chemotherapy until the date of RECIST v1.1 (which was defined as imaging disease progression or death). The objective response rate (ORR) was defined as the number of patients with a complete response (CR) plus the number of patients with a partial response (PR) divided by the total number of patients. The disease control rate (DCR) was defined as the number of patients with CR, PR, and stable disease (SD) divided by the total number of patients. The AEs were graded according to the National Cancer Institute Common Terminology Criteria for Adverse Events, version 5.0.

### 2.4. Statistical Analysis

Continuous variables are presented as the means and standard deviations, and the categorical variables as frequencies and percentages. Binary variables were compared using Fisher’s exact test, and continuous variables were compared using the Mann–Whitney U or Kruskal–Wallis tests. Survival curves were generated, and a log-rank test was performed. Statistical significance was set at *p* < 0.05. All the statistical analyses were performed using SPSS version 29 (IBM Corp., Armonk, NY, USA).

## 3. Results

### 3.1. Patient Characteristics

Patient characteristics are shown in Table 1. A total of 44 patients were included in this study, comprising 21 men and 23 women with a median age of 65 years (range 46–82 years). The ECOG PS score was 0 in 35 of the patients and 1 in 9 of the patients. There were no cases of PS 2. The primary BTC lesions were gallbladder cancer in 13, intrahepatic bile duct cancer in 19, extrahepatic bile duct cancer in 11, and ampullary carcinoma in 1 patients. Microsatellite instability was positive for 1 patient among the 10 patients tested. Among the 44 cases, 5 were locally advanced, 18 were distant metastases, and 21 were postoperative recurrences. Tumor markers at the start of treatment were median Carcinoembryonic antigens at 5.2 ng/mL (range of <2.0–349) and median Carbohydrate antigens at 19-9 56.7 ng/mL (range of <2.0–72,862). Durvalumab administration time was median 7.5 (range times of 1–22). In the limited primary chemotherapy group, the median administration time was 9 (range times 1–22) (Table 2). After eight courses of GCD administration, 20 patients (45.5%) transitioned to durvalumab maintenance chemotherapy. In six of the cases, chemotherapy was discontinued due to death or PD before the completion of eight courses of GCD chemotherapy. A total of 19 patients (43.2%) required bile duct drainage because of obstructive jaundice or biliary infection during the course of the disease. Treatment was introduced primarily in 30 cases (68.4%), and it was introduced secondarily and beyond in 14 patients. Cisplatin was administered at a reduced dose from the start of administration in two patients due to renal dysfunction and in seven patients who had received cisplatin prior to GCD administration, respectively.

### 3.2. Overall Survival

Table 2, Table 3 and Table 4 and Figure 2 and Figure 3 demonstrate the OS results. The overall median OS was 15.3 months (95% confidence interval (CI): 9.8–20.7). In the comparative survival analysis between the two cohorts (primary chemotherapy group vs. subsequent chemotherapy groups), a statistically significant difference (*p* = 0.014) was observed despite the primary chemotherapy group maintaining a survival probability greater than 0.5. The mean OS of the group treated with the primary therapy was 16.8 months (95% CI; 13.7–19.8). Furthermore, the group requiring bile duct drainage exhibited inferior OS compared to those who did not (*p* = 0.003). When comparing locally advanced, postoperative recurrent, and metastatic cases, the non-metastatic group maintained a survival probability greater than 0.5, with no statistical difference (*p* = 0.47). The OS among ECOG PS 0 and 1 cases was 12.4 months (95% CI; 7.3–17.5) and 15.9 months (95% CI: 7.8–24.0), respectively (*p* = 0.21).

### 3.3. Progression-Free Survival

As shown in Table 2, Table 3 and Table 4 and Figure 2 and Figure 3, the overall median PFS was 8.0 months (95% CI: 5.2–10.9). For patients treated with primary chemotherapy versus those receiving subsequent chemotherapies, median PFS was 8.7 months (95% CI: 3.0–14.4) and 3.0 months (95% CI: 2.0–4.0), respectively (*p* = 0.076). The group requiring drainage had a significantly worse PFS (15.1 months vs. 5.0 months, *p* = 0.029). The PFS among locally advanced or postoperative recurrent and metastatic cases was 12.7 months (95% CI: 4.3–21.0) and 6.5 months (95% CI: 2.3–10.7), respectively (*p* = 0.062). When comparing ECOG PS 0 and 1, there was no difference (*p* = 0.25).

### 3.4. Clinical Response

Table 2, Table 4, Table 5 and Table 6 shows the clinical responses of the patients. The clinical response was CR in 3, PR in 7, SD in 26, and PD in 8. The overall clinical response rate was 0.23 (10/44) in ORR and 0.82 (36/44) in DCR. Limited to the primary chemotherapy group, among 30 cases, there was CR in 3, PR in 6, SD in 17, and PD in 4. ORR was 0.30 (9/30), and DCR was 0.87 (26/30) (Table 2 and Table 3). When comparing the primary chemotherapy group and subsequent chemotherapy group, the ORR was 0.30 vs. 0.07 (*p* = 0.096) and the DCR was 0.87 vs. 0.71 (*p* = 0.23) (Table 5). The ORR had no statistically significant elements in either univariate or multivariate analyses. In contrast, the absence of metastasis was a factor in achieving an objective response in the univariate analysis (*p* = 0.047). In multivariate analyses, the DCR tended to be better in patients without metastases, although the difference was not statistically significant (*p* = 0.06, odds ratio (OR): 5.31, 95%CI: 0.93–30.2) (Table 6). Conversion surgery was performed in one case of intrahepatic cholangiocarcinoma that became PR. In 11 of the 20 patients (55%) who were transitioned to durvalumab maintenance chemotherapy, the disease was under control, and long-term survival was achieved with continued chemotherapy.

### 3.5. Adverse Events

Table 7 shows the AEs and im-AEs. A total of 24 cases (54.5%) developed Grade 3 or more AEs with duplication: with neutropenia in 13, biliary tract infection in 7, anemia in 6, thrombocytopenia in 3, and other in 5 patients. Biliary tract infections were treated with antibiotics, biliary drainage, or both, and blood cell depletion was treated with blood transfusion, drug withdrawal, dose reduction, and administration of the granulocyte colony-stimulating factor. No deaths occurred due to AEs. In two cases (4.6%), im-AEs developed, including an infusion reaction and hypothyroidism in one case each. Both patients were treated with drugs, and chemotherapy was continued.

## 4. Discussion

This study assessed the safety and effectiveness of GCD chemotherapy for treating BTCs. Traditionally, BTCs have poor response to chemotherapy, with few regimens expected to improve the prognosis. Since a clinical trial in 2010 reported an OS advantage of GC chemotherapy over gemcitabine for advanced BTCs, GC therapy has been the first choice for advanced or recurrent BTCs [6]. GC therapy has been the prevailing standard for first line therapy, achieving median OS figures of 11.5–11.7 months [6,7,8,9]. A phase III trial comparing GC with GS chemotherapy showed non-inferiority of GS to GC therapy but did not prove superiority, and it also did not show significant differences in response rates (OS: 15.1 months vs. 13.4 months, HR: 0.945, 90%CI: 0.777–0.1.149) [3]. The other phase III trials, which compared GC and GCS therapy, also demonstrated the superiority of GCS therapy over GC therapy (OS: 13.5 months vs. 12.6 months, HR: 0.79, 90% CI: 0.628–0.966), and it became the new standard of care for BTCs. The response rates were 15% and 41.5% for GCS chemotherapy [4]. Recently, the ABC-06 trial compared a symptom control group with the addition of modified folinic acid, fluorouracil, and oxaliplatin (mFOLFOX) in patients with advanced BTCs who had progressed after primary treatment with GC chemotherapy. In this trial, mFOLFOX chemotherapy improved OS for 5.3 months to 6.2 months when compared to the symptom control group, and it was approved overseas as a second line treatment [10].

The efficacy of GCD chemotherapy was reported in the Topaz-1 trial in 2022 for BTC patients in primary chemotherapy. In the Topaz-1 trial, the median duration of follow-up was 16.8 months (95% CI, 14.8 to 17.7). The median OS was 12.8 months (95% CI, 11.1 to 14.0), the median PFS was 7.2 months (95% CI, 6.7 to 7.4), and the ORR was 0.27 [5]. In previous reports of real-world data from the Topaz-1 trial onward, the median PFS was 5.1, 8.3, and 8.9 months, and the median OS was 10.3, 12.9 and 14.8 months, respectively [11,12,13]. In the present study, the median duration of follow-up was shorter than that in previous reports; however, the OS and PFS were longer than the equivalent in previous reports, including those outside the primary chemotherapy group included in the Topaz-1 trial [5,11,12,13]. GCD chemotherapy was considered an effective regimen with high ORR and DCR. PFS and OS were longer in the primary chemotherapy group than in the control group. In secondary and subsequent cases, patients may have poor general and nutritional conditions due to previous chemotherapy. Dose reduction or discontinuation of the drug may be necessary because of its side effects and cumulative toxicity of cisplatin [13]. In the present study, cisplatin was reduced from the starting dose in nine patients who received GCD in secondary and beyond chemotherapy due to impaired renal function or the cumulative toxicity of cisplatin. It is important to introduce GCD chemotherapy as primary chemotherapy whenever possible. Unlike the Topaz-1 trial, which even included patients with PS 2, this study included only patients in good general condition with ECOG PS of 0 or 1, and there was no difference in treatment outcome according to the PS score [5]. However, in general, patients with poor ECOG PS, as well as those with a history of chemotherapy, may have difficulty initiating and continuing chemotherapy due to poor nutritional and general health status.

These findings suggest that GCD therapy is effective, with an ORR comparable to that observed in the Topaz-1 trial. The analysis highlighted that the presence of metastasis significantly influenced therapeutic responses, emphasizing the potential advantages of GCD chemotherapy in non-metastatic cases. The smaller tumor volume in cases of postoperative recurrence and locally advanced disease may be the reason for the better prognosis compared to metastatic cases. Although the non-metastatic cases had longer PFS than metastatic cases, the lack of statistical significance could be attributed to the limited sample size and follow-up duration. The longer PFS and OS in this study compared to the Topaz-1 study may be because more than 80% of the cases in the Topaz-1 study were metastatic compared to the 40% in our study [5]. Metastatic cases may have a higher tumor volume than locally advanced or postoperative recurrent cases, and they may have an inferior response rate.

Sometimes, chemotherapy for BTCs is a battle against obstructive jaundice due to the tumor growth or biliary infection caused by stent dysfunction. Continuing chemotherapy without cholangitis or obstructive jaundice due to stent dysfunction is critical to prolonging the patient’s prognosis. In particular, the endoscopic drainage of hilar cholangiocarcinoma is a complex and unestablished procedure based on its anatomical characteristics. As a result, cholangitis management is often difficult [14,15]. In the past, drainage for hilar cholangiocarcinoma of at least 25% of the total liver volume was required, and the usefulness of single lobe drainage was argued because of fewer AEs and a similar duration of patency [16,17,18]. However, the drainage of at least 50% of the total volume has been shown to contribute more to long-term survival and successful improvement of jaundice than the drainage of less than 50%, and more than 50% is now recommended [19,20]. With the advent of GCD chemotherapy, the long-term prognosis of cholangiocarcinoma has become promising and, recently, the usefulness of drainage with plastic stents for re-intervention, easy stent replacement, and long-term patency has been reported. The success rate and patency rate of bilateral inside stent (IS) are superior to those of single lobe IS, and the usefulness of bilateral IS for re-intervention has been reported. It is expected that plastic bilateral IS will be replaced with metal stents [21,22]. There is still insufficient evidence for biliary drainage by bilateral IS in both lobes, and it is desirable to accumulate evidence on appropriate drainage methods that cause less interruptions of chemotherapy. Patients with bile duct stenting for obstructive jaundice underwent endoscopic treatment and bile duct stent replacement at around every three months before the onset of cholangitis. Even if cholangitis developed, drainage was performed as early as possible to shorten the duration of antimicrobials and the interruption of chemotherapy. We usually place bilateral IS for hilar cholangiocarcinoma because of the usefulness of bilateral IS for re-intervention. Endoscopic transpapillary drainage is sometimes difficult due to tumor progression; in such cases, drainage is also aggressively performed using ultrasound endoscopic drainage. In addition, we used intraprocedural hologram support with the mixed-reality technique to place the stent in the appropriate bile duct branch [23].

Treatment of biliary infections also includes antimicrobial therapy. It has been reported that the OS is shortened when antimicrobials are administered to immune checkpoint inhibitors (ICI)-treated patients because of changes in the intestinal microflora [24]. In the present study, patients who required bile duct drainage were also more likely to have treatment interruptions or delays in their starting treatment because of the antimicrobials used to treat cholangitis or endoscopic treatment. In our cohort, patients requiring bile duct drainage tended to have poorer OS and PFS outcomes, highlighting the need for effective strategies to manage these complications and reduce treatment interruptions.

In the Topaz-1 study, Grade 3 or higher AEs were observed in 75.7% of patients and im-AEs in 2.4% [5]. Compared with the outcomes reported by the Topaz-1 trial, our study observed a lower incidence of Grade 3 or higher AEs, suggesting a favorable safety profile within our population. However, the incidence of im-AEs was similar, indicating the need for vigilant monitoring throughout the treatment course. In this study, Grade 3 or higher AEs had fewer and similar levels of im-AEs when compared with the Topaz-1 study. As im-AEs are sometimes severe, additional periodic endocrine function tests should be performed to diagnose early and consult a specialist before therapeutic intervention. It is also important to establish a system to deal with AEs in a multidisciplinary manner.

In recent years, advances in cancer genome profiling have led to regimen selection based on genetic mutations; however, few drugs can be used clinically in BTCs, although genetic mutations are more common than in other gastrointestinal cancers [2]. In BTCs, depending on the report, 25–55% of the gene mutations are reported to be targeted, and therapies have been developed for several important genomic mutations [25,26]. FGFR-2 inhibitors have recently been shown to be effective against FGFR fusion gene mutations and are now available; however, the actual detection rate is not high, and their use in daily practice is limited [27,28]. IDH 1/2 gain-of-function mutations, or B-Raf proto-oncogenes, are also important therapeutic targets for BTCs. Various clinical studies are underway for these genetic mutations, and it is expected that there will be even more treatment options in the future [29]. Currently, ICIs are used alone or in combination with chemotherapy for several gastrointestinal cancers, including esophageal and gastric cancer. In previous reports on the efficacy study of the administration of ICIs for BTCs, the response rate was 3.3–22%, the median PFS was 1.4–3.7 months, and the OS was 5.2–14.2 months. However, no effective treatments have yet emerged. The Topaz-1 trial has also been conducted, but there is no certainty whether the expression of programmed cell death-ligand 1 is a predictor of efficacy [29,30,31]. However, this was not investigated in the present study, and further studies are required. As mentioned earlier, biliary tract cancers are more likely than malignant tumors of other organs to have genetic mutations that are potentially amenable to treatment, but it is difficult to provide chemotherapy that matches the genetic mutations in actual clinical practice [5,13].

GCD chemotherapy is considered an effective regimen with regard to the prolongation of OS and PFS when compared with conventional regimens, such as GC or GS chemotherapy [3,4,5,6,7,8,9,10,11,12,13,14]. Therefore, GCD chemotherapy is the first line of treatment for the time being. Future studies should investigate whether the prognosis can be prolonged after patients are transferred to the administration of durvalumab after completing eight courses of GC and durvalumab combined therapy. In this study, more than half of the patients achieved disease control and long-term survival after conversion to durvalumab maintenance therapy. Studies on subsequent regimens after GCD chemotherapy are limited. In Japan, mFOLFOX therapy for BTCs is not covered by insurance and cannot be used as a second line therapy. In our institution, if a patient discontinues treatment due to clinical or imaging (per RECIST v1.1) disease progression while on GCD chemotherapy, the patient is switched to GS therapy or S-1 monotherapy. Cisplatin combination therapy may be difficult to administer for a long period of time due to nephrotoxicity and other problems, but if the patient is already on maintenance therapy in durvalumab, we will switch to GC, GS therapy, or S-1 monotherapy [13]. There is no clearly defined recommended regimen for the secondary chemotherapy after GCD therapy [5,13]. Further evidence is required through the accumulation of more cases in the future.

To the best of our knowledge, this is the first study of GCD chemotherapy for BTCs in Japan. GCD chemotherapy is considered an effective chemotherapy for BTCs; however, due to its early use, the number of cases was small, the observation period was shorter, and many cases were not transferred to durvalumab maintenance therapy. Further long-term observations and an accumulation of more cases are needed. The present study showed inferior outcomes in cases requiring biliary drainage and in cases in which GCD chemotherapy was administered as the secondary and beyond chemotherapy.

This study has a few limitations, including its retrospective nature, single institution setting, small sample size, and shorter observation period compared with earlier studies. These limitations may affect the findings’ generalizability and robustness, highlighting the need for further research with extended follow-up periods and larger, more diverse patient cohorts.

## 5. Conclusions

In conclusion, GCD chemotherapy is a promising option for BTCs, with encouraging efficacy and safety profiles. Expanded, prospective studies are required to confirm its role as the preferred primary treatment and to investigate potential subsequent treatment strategies following GCD therapy.

## Figures and Tables

**Figure 1 cancers-17-00314-f001:**
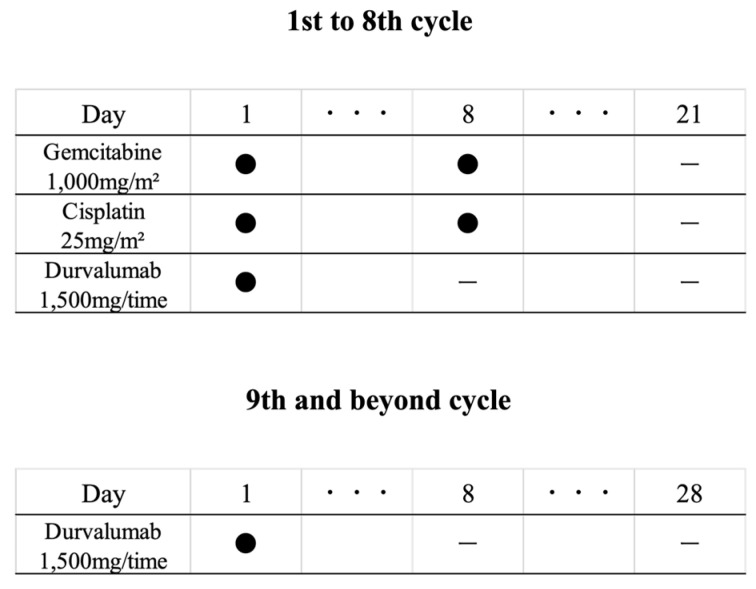
Durvalumab plus gemcitabine and cisplatin (GCD) chemotherapy administration schedule.

**Figure 2 cancers-17-00314-f002:**
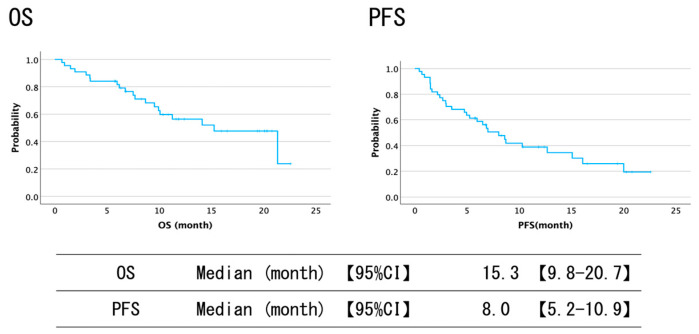
Survival curves of durvalumab plus gemcitabine and cisplatin chemotherapy. Survival curves of the overall results following durvalumab plus gemcitabine and cisplatin administration. The median overall survival (OS) was 15.3 months (95% confidence interval (CI): 9.8–20.7), and the median progression-free survival (PFS) was 8.0 months (95% CI: 5.2–10.9).

**Figure 3 cancers-17-00314-f003:**
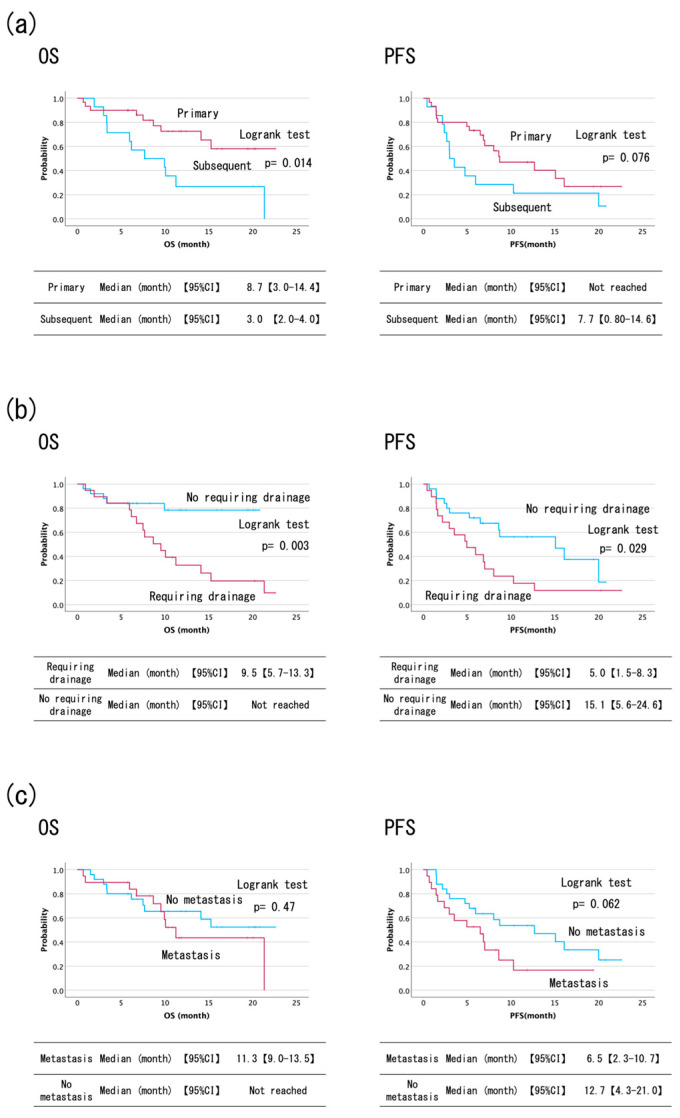
Subgroup analysis of the survival curves of durvalumab plus gemcitabine and cisplatin chemotherapy. (**a**) Survival curves of durvalumab plus gemcitabine and cisplatin according to primary chemotherapy. The probability of OS in the primary therapy group was greater than 0.5. The primary chemotherapy group exhibited a significantly longer OS than the second and subsequent chemotherapy groups (*p* = 0.014). For patients treated with primary chemotherapy, the median PFS was 8.7 months, whereas it was 3.0 months for those who received subsequent therapies (*p* = 0.076). (**b**) Survival curves of durvalumab plus gemcitabine and cisplatin by requirement for biliary drainage. The group requiring bile duct drainage demonstrated inferior OS compared to those who did not require drainage (*p* = 0.003). Additionally, the group requiring drainage had a significantly worse PFS (*p* = 0.029). (**c**) Survival curves of durvalumab plus gemcitabine and cisplatin according to metastasis status. When comparing locally advanced cases, postoperative recurrent cases, and metastatic cases, the probability of OS in non-metastatic cases was greater than 0.5, and there was no statistical difference observed (*p* = 0.47). Furthermore, there was no statistically significant difference in PFS among the locally advanced, postoperative recurrent, and metastatic cases (*p* = 0.062).

**Table 1 cancers-17-00314-t001:** Patient characteristics.

		n = 44
Age (yrs)	Median (year) [range]	65 [46–82]
Sex	Male	21
	Female	23
PS	0	35
	1	9
Chemotherapy line	Primary	30
	Subsequent	14
Biliary drainage		25
Primary lesions	Intrahepatic bile duct cancer	19
	Gallbladder cancer	13
	Extrahepatic bile duct cancer	11
	Ampullary carcinoma	1
Tumor marker (At the start of administration)	CEA (ng/mL) [range]	5.2 [<2.0–349]
	CA19-9 (ng/mL) [range]	56.7 [<2.0–72,862]
Diesease status	Postoperative recurrence	21
	Metasasis	18
	Locally advance	5

CA, Carbohydrate antigen; CEA, Carcinoembryonic antigen; PS, Performance status.

**Table 2 cancers-17-00314-t002:** Chemotherapy results.

		Overall	Primary Chemotherapy	Subsequent Chemotherapy
Clinical response	CR	2	2	0
	PR	8	7	1
	SD	26	17	9
	PD	8	4	4
	ORR	0.23	0.30	0.07
	DCR	0.82	0.87	0.71
OS	Median (month) [95%CI]	15.3 [9.8–20.7]	not reached	7.7 [0.80–14.4]
PFS	Median (month) [95%CI]	8.0 [5.2–10.9]	8.7 [3.0–14.4]	3.0 [2.0–4.0]
Durvalumab administration time	Median (time) [range]	7.5 [1–22]	9 [1–22]	4 [1–22]

CI, confidence interval; CR, Complete response; DCR, disease control rate; ORR, objective response rate; OS, Overall survivals; PD, Progressive disease; PFS, Progression free survivals; PR, Partial response; SD, Stable disease.

**Table 3 cancers-17-00314-t003:** Comparison of the OS and PFS between the two groups.

		OS (month) [95%CI]	PFS (month) [95%CI]
Drinage	Requiring drainage	9.5 [5.7–13.3]	5.0 [1.5–8.3]
	No requiring drainage	not reached	15.1 [5.6–24.6]
	*p*-value	0.003	0.029
Chemotherapy	Primary chemotherapy	not reached	8.7 [3.0–14.4]
	Subsequent chemotherapy	7.7 [0.80–14.6]	3.0 [2.0–4.0]
	*p*-value	0.014	0.076
PS	0	12.4 [7.3–17.5]	16.5 [9.3–23.7]
	1	15.9 [7.8–24.0]	not reached
	*p*-value	0.21	0.25
Disease status	Metastasis	11.3 [9.0–13.5]	6.5 [2.3–10.7]
	No metastasis	not reached	12.7 [4.3–21.0]
	*p*-value	0.47	0.062

CI, confidence interval; OS, Overall survivals; PFS, Progression free survivals; PS, Performance status.

**Table 4 cancers-17-00314-t004:** Comparison of the chemotherapy results between the Topaz-1 trial and present study.

		Topaz-1 trial [5]	Shionoya, et al.
Parameter			
The number of patients no.		341	44
Age (yrs)	Median (year) [range]	64 [20–84]	65 [46–82]
Sex	Male no.	169	21
	Female no.	172	23
Median follow up period (months)		16.8	10
Diesease status	Postoperative recurrence no. (%)	67 (19.6)	21 (47.4)
	Metasasis no. (%)	303 (88.9)	18 (40.9)
	Locally advance no. (%)	38 (11.1)	5 (11.4)
Primary lesions	Intrahepatic bile duct cancer	190 (55.7)	19 (43.2)
	Gallbladder cancer	85 (24.9)	13 (29.5)
	Extrahepatic bile duct cancer	66 (19.4)	11 (25)
	Ampullary carcinoma	0 (0)	1 (2.3)
Response rate	ORR	0.27	0.23
	DCR	0.85	0.82
OS Median (month) [95%CI]		12.8 [11.1–14.0]	15.3 [9.8–20.7]
PFS Median (month) [95%CI]		7.2 [6.7–7.4]	8.0 [5.2–10.9]
Grade 3 or higher AE (%)		75.7	54.5
Immune-mediated-AE (%)		2.4	4.5

AE, Adverse events; CI, confidence interval; DCR, disease control rate; ORR, objective response rate; OS, Overall survivals; PFS, Progression free survivals.

**Table 5 cancers-17-00314-t005:** The clinical response rate of durvalumab plus gemcitabine and cisplatin administration.

		ORR	DCR
Overall results		0.23 (10/44)	0.82 (36/44)
Age	65 year>	0.26 (6/23)	0.78 (18/23)
	65 year≤	0.19 (4/21)	0.86 (18/21)
	*p*-value	0.58	0.53
Sex	Male	0.24 (5/21)	0.76 (16/21)
	Female	0.22 (5/23)	0.87 (20/23)
	*p*-value	0.87	0.36
PS	0	0.26 (9/35)	0.83 (29/35)
	1	0.11 (1/9)	0.78 (7/9)
	*p*-value	0.51	0.82
Chemotherapy line	Primary chemotherapy	0.30 (9/30)	0.87 (26/30)
	Subsequent chemotherapy	0.07 (1/14)	0.71 (10/14)
	*p*-value	0.096	0.23
Biliary drainage	Requiring drainage	0.21 (4/19)	0.79 (15/19)
	No requiring drainage	0.24 (6/25)	0.84 (21/25)
	*p*-value	0.82	0.67
Disease state	Metastasis	0.32 (6/19)	0.68 (13/19)
	No metastasis	0.16 (4/25)	0.92 (23/25)
	*p*-value	0.23	0.047

DCR, disease control rate; ORR, objective response rate; PS, performance status.

**Table 6 cancers-17-00314-t006:** Statical analysis on clinical response.

	Univariate Analysis	Multivariate Analysis		
ORR	*p*-Value	*p*-Value	Odds Ratio	95% CI
65 years>	0.58	0.58		
Male	0.87	0.87		
PS	0.51	0.35		
Primary chemotherapy	0.10	0.09		
Requiring drainage	0.82	0.82		
No metastasis	0.23	0.22		
	Univariate analysis	Multivariate analysis		
**DCR**	***p*-value**	***p*-value**		
65 years>	0.53	0.29		
Male	0.36	0.10		
PS	0.82	0.35		
Primary chemotherapy	0.23	0.12		
Requiring drainage	0.67	0.92		
No metastasis	0.047	0.06	5.31	0.93–30.2

CI, confidence interval; DCR, disease control rate; ORR, objective response rate; PS, performance status.

**Table 7 cancers-17-00314-t007:** Adverse events.

Adverse events (≥Grade 3)	
Neutropenia	13
Biliary tract infection	7
Anemia	6
thrombocytopenia	3
Others	5
Immune-mediated Adverse events	
infusion reaction	1
hypothyroidism	1

## Data Availability

The data presented in this study are available on request from the corresponding authors. The data are not publicly available due to the ethics approval agreement.

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
