# Peer review of "Initial Use Experience of Durvalumab Plus Gemcitabine and Cisplatin for Advanced Biliary Tract Cancer in a Japanese Territory Center"

_cancers, 2025, doi:10.3390/cancers17020314_

Round 1
Reviewer 1 Report
Comments and Suggestions for Authors
The manuscript of Shionoya et al. entitled "Initial Use Experience of Durvalumab Plus Gemcitabine and Cisplatin for Advanced Biliary Tract Cancer in a Japanese Territory Center" addresses an important and timely topic in oncology by evaluating the efficacy and safety of GCD (gemcitabine, cisplatin, durvalumab) chemotherapy for biliary tract cancers (BTCs). In general the paper is well-written and the subject can be of interest for Cancers MDPI readers. However, in order to improve paper quality, some improvements suggestions are addressed:
(1) The introduction effectively highlights the challenges of managing BTCs and the significance of immune checkpoint inhibitors like durvalumab in improving treatment outcomes. However, it does not adequately articulate the novelty of this study compared to existing research, such as the Topaz-1 trial. The authors should emphasize how this study advances the understanding of GCD therapy, particularly in the context of Japanese patient populations or specific clinical practices like bile duct management. Additionally, certain claims, such as the limited availability of second-line treatments and the challenges of biliary drainage, require stronger referencing to recent and relevant literature.
(2) The methodology is described but would benefit from additional detail to improve reproducibility. The inclusion and exclusion criteria should be more explicitly defined, particularly for differentiating primary and subsequent chemotherapy groups. The administration protocol for GCD therapy needs elaboration, including details on dose adjustments, patient monitoring, and management of AEs. A schematic diagram summarizing the study design, patient cohorts, and treatment timelines would provide clarity and enhance understanding.
(3) The results are presented in a structured manner, with clear reporting of survival outcomes, response rates, and adverse events. However, the analysis could be expanded to explore the clinical implications of the findings. For instance, the survival benefit in non-metastatic versus metastatic cases warrants deeper discussion of tumor biology and treatment response. Similarly, the poorer outcomes in subsequent chemotherapy groups should be analyzed in relation to factors like baseline performance status or cumulative toxicity. The survival curves in Figure 1 are informative but require better annotation, including clear legends and consistent axis labeling.
(4) The discussion reiterates the findings but lacks a critical analysis of their broader implications. Comparing the study outcomes with those of the Topaz-1 trial and similar real-world studies would contextualize the results and strengthen their contribution to the field. The role of bile duct management in improving survival outcomes is noted but not explored in sufficient depth. Specific strategies, such as optimizing drainage techniques to reduce chemotherapy interruptions, should be discussed. While the limitations of the study, including its retrospective nature and small sample size, are acknowledged, the authors could propose strategies for addressing these issues in future research.
(5) The figures and tables included in the manuscript are helpful but require refinement. Ensuring high-resolution images, clear labeling, and consistent formatting would improve their presentation. Adding a summary table comparing this study's findings with the Topaz-1 trial would provide valuable context.
Author Response
Comments 1: The introduction effectively highlights the challenges of managing BTCs and the significance of immune checkpoint inhibitors like durvalumab in improving treatment outcomes. However, it does not adequately articulate the novelty of this study compared to existing research, such as the Topaz-1 trial. The authors should emphasize how this study advances the understanding of GCD therapy, particularly in the context of Japanese patient populations or specific clinical practices like bile duct management. Additionally, certain claims, such as the limited availability of second-line treatments and the challenges of biliary drainage, require stronger referencing to recent and relevant literature.
Response 1: Thank you for your comments.
We consider the novelty of this article is that there is a discussion of the 2nd line and beyond and biliary drainage for biliary tract cancer patients, which were not included in the Topaz-1 study. We have added an overall description of these issues (p8 259-262, p 10 343-346, 361-362). We have noted difference by performance status. ( p 4 156-157, p 6, 194-195, p 8 263-p9 268 )
Since we had already mentioned the treatment outcome comparing metastatic and non-metastatic cases, we added details about metastatic cases having a worse treatment outcome (p 9 278-280). We also added that secondary treatment is limited and that biliary drainage is difficult, including how to deal with this problem in our hospital (p9 299-308, p368-370).
Comments 2:
The methodology is described but would benefit from additional detail to improve reproducibility. The inclusion and exclusion criteria should be more explicitly defined, particularly for differentiating primary and subsequent chemotherapy groups. The administration protocol for GCD therapy needs elaboration, including details on dose adjustments, patient monitoring, and management of AEs. A schematic diagram summarizing the study design, patient cohorts, and treatment timelines would provide clarity and enhance understanding.
Response 2:
Thank you for your important comments.
As your comments, we added the inclusion and exclusion criteria for this study. We also described the method of patient monitoring (p2 87-94)and created a figure for the dosing schedule (Figure 1). Table 1 and 4 were also showed patients’ characteristics’. We modified inclusion and exclusion criteria (p2 76-79).
Comments 3:
The results are presented in a structured manner, with clear reporting of survival outcomes, response rates, and adverse events. However, the analysis could be expanded to explore the clinical implications of the findings. For instance, the survival benefit in non-metastatic versus metastatic cases warrants deeper discussion of tumor biology and treatment response. Similarly, the poorer outcomes in subsequent chemotherapy groups should be analyzed in relation to factors like baseline performance status or cumulative toxicity. The survival curves in Figure 1 are informative but require better annotation, including clear legends and consistent axis labeling.
Response 3:
Thank you for your kind comments.
We had already discussed the difference in treatment outcome between non-metastatic and metastatic cases, so we added a postscript and an additional discussion of the difference in treatment outcome due to performance status and cumulative toxicity (p4 141-143, p8 259- p9 268).
In addition, we have modified survival curves in two parts and revised (Figure 2 and 3).
Comments 4:
The discussion reiterates the findings but lacks a critical analysis of their broader implications. Comparing the study outcomes with those of the Topaz-1 trial and similar real-world studies would contextualize the results and strengthen their contribution to the field. The role of bile duct management in improving survival outcomes is noted but not explored in sufficient depth. Specific strategies, such as optimizing drainage techniques to reduce chemotherapy interruptions, should be discussed. While the limitations of the study, including its retrospective nature and small sample size, are acknowledged, the authors could propose strategies for addressing these issues in future research.
Response 4:
Thank you for your important comments.
We have additionally described the ingenuity of bile duct drainage (p9 299-307).
Comments 5:
The figures and tables included in the manuscript are helpful but require refinement. Ensuring high-resolution images, clear labeling, and consistent formatting would improve their presentation. Adding a summary table comparing this study's findings with the Topaz-1 trial would provide valuable context.
Response 5:
Thank you for your kind comments.
We have revised the overall survival curves by changing the font style. We also added details in the figure legend of Figure 2 .
Also, we made a table to compare Topaz-1 vs. our study (Table 4).
Reviewer 2 Report
Comments and Suggestions for Authors
In this manuscript, the authors evaluated the safety and effectiveness of GCD therapy for treating BTC in Japan. Results showed an ORR of 23% and a DCR of 82%. Median OS and PFS were 15.3 and 8.0 months, respectively. Patients receiving primary chemotherapy exhibited prolonged survival, while bile duct drainage was associated with improved outcomes. Notably, grade 3 or higher AEs occurred in 54.5% of patients. Overall, GCD treatment demonstrates significant potential for BTC treatment, underscoring the importance of bile duct management in enhancing patient outcomes. The manuscript has some limitations, while it is helpful in clinical settings when determining which BTC patients may benefit from GCD treatment. Overall, there are still two issues to be addressed before considering its acceptance.
Major comments:
1. DOR (duration of response) is one of critical indicators for assessing the effectiveness of GCD therapy. Therefore, the authors should show the related information in the result section and provide the comparison with other chemotherapy regimens in the discussion section.
2. GCD therapy is a promising option for BTC with encouraging efficacy and safety profiles, particularly in primary therapy or no metastatic cases. Is there any serious TRAE (treatment-related adverse events) of GCD therapy distinct from traditional chemotherapy? The authors should provide further discussions in the manuscript.
Author Response
Comments 1:
DOR (duration of response) is one of critical indicators for assessing the effectiveness of GCD therapy. Therefore, the authors should show the related information in the result section and provide the comparison with other chemotherapy regimens in the discussion section.
Response 1:
Thank you for your comment.
We already discussed about other chemotherapy results in the discussion section (p 8 228-245, p10 347-363). We added a postscript and an additional discussion.
Commets 2:
GCD therapy is a promising option for BTC with encouraging efficacy and safety profiles, particularly in primary therapy or no metastatic cases. Is there any serious TRAE (treatment-related adverse events) of GCD therapy distinct from traditional chemotherapy? The authors should provide further discussions in the manuscript.
Response 2:
Thank you for your important comment.
As your comments, we added discussion about AE and im-AE in the discussion section and our protocol about AE monitoring. (p2 87- p3 94, p 10 322-325)
Round 2
Reviewer 2 Report
Comments and Suggestions for Authors
Following the authors’ responses, the revised manuscript has been sufficiently improved to warrant publication in Cancers.